# Mechanism of sensor kinase CitA transmembrane signaling

Xizhou Cecily Zhang[1], Kai Xue [1], Michele Salvi[1], Benjamin Schomburg[1], Jonas Mehrens[1], Karin Giller[1], Marius Stopp[2], Siegfried Weisenburger [3,4], Daniel Böning[3,4], Vahid Sandoghdar [3,4], Gottfried Unden[2], Stefan Becker [1]✉, Loren B. Andreas [1]✉ & Christian Griesinger [1]✉

Membrane bound histidine kinases (HKs) are ubiquitous sensors of extracellular stimuli in bacteria. However, a uniform structural model is still missing for their transmembrane signaling mechanism. Here, we used solid-state NMR in conjunction with crystallography, solution NMR and distance measurements to investigate the transmembrane signaling mechanism of a paradigmatic citrate sensing membrane embedded HK, CitA. Citrate binding in the sensory extracytoplasmic PAS domain (PASp) causes the linker to transmembrane helix 2 (TM2) to adopt a helical conformation. This triggers a piston-like pulling of TM2 and a quaternary structure rearrangement in the cytosolic PAS domain (PASc). Crystal structures of PASc reveal both anti-parallel and parallel dimer conformations. An anti-parallel to parallel transition upon citrate binding agrees with interdimer distances measured in the lipid embedded protein using a site-specific $^{19}F$ label in PASc. These data show how Angstrom scale structural changes in the sensor domain are transmitted across the membrane to be converted and amplified into a nm scale shift in the linker to the phosphorylation subdomain of the kinase.

In microorganisms, membrane receptors are essential for processing extracellular stimuli. As a part of the ubiquitous two-component signaling systems (TCS), membrane-bound sensor histidine kinases (HKs) are one of the most abundant classes of prokaryotic membrane receptors. Many bacteria contain dozens to hundreds of sensors for perceiving various environmental stimuli[1]. The transmembrane signaling pathway in extra-cellular sensing TCSs follows a common scheme: the initial signal that switches the HK from the ligand-free state to the ligand-bound state is generated by cue perception in the sensor domain, which, through a transmembrane (TM) domain and cytoplasmic Per-Arnt-Sim (PAS), or HAMP (named after the proteins Histidine kinases, Adenylate cyclases, Methyl accepting proteins and Phosphatases) domain, leads to cross-phosphorylation in the dimeric kinase core at a histidine residue (DHp). Subsequently, the kinase domain phosphorylates the respective response regulator (RR) protein, resulting ultimately in the regulation of target gene(s). Transmembrane (TM) signaling mechanisms were postulated, based on the structures of the isolated extracytoplasmic receiver domain for KinB[2], TorT/TorS[3], LuxPQ[4], CitA[5,6], NarX[7], and full-length DcuS[8,9]. The structure of the complete cytoplasmic region has been solved for some proteins, such as VicK, YF1 and HK853[10–12]. However, direct structural insight into the transmembrane signaling process is needed for PASc containing HKs.

The citrate sensor HK (CitA) is an attractive HK to study, because it is functional without any auxiliary proteins[13]. It is a member of the PAS domain containing HKs which comprise 30% of all HKs[14]. CitA belongs to a sensor kinase subfamily[15] of HKs, which can bind to citrate or C4-dicarboxylates. This class of sensor HK has two PAS domains, a

[1]NMR-based Structural Biology, Max Planck Institute for Multidisciplinary Sciences, Göttingen, Germany. [2]Institute for Molecular Physiology (imP), Microbiology and Biotechnology, Johannes Gutenberg University, Mainz, Germany. [3]Department of Physics, Friedrich Alexander University (FAU) Erlangen-Nürnberg, Erlangen, Germany. [4]Department Nano-Optics, Plasmonics and Biophotonics, Max Planck Institute for the Science of Light, Erlangen, Germany. ✉e-mail: sabe@mpinat.mpg.de; land@mpinat.mpg.de; cigr@mpinat.mpg.de

periplasmic (or extracytoplasmic) ligand binding domain (PASp) and a cytoplasmic transmitter domain (PASc). The fold of PASp has also been named PDC domain (PhoQ/DcuS/CitA protein family) due to the specific features of its fold[7]. In accordance with the naming of the PASc domain, the periplasmic domain is termed PASp here. The PASp and PASc domains are connected to the TM2 helix by two linkers: the PASp/TM2-linker and the TM2/PASc-linker (topology and domain organization shown in Fig. 1A, B, and Supplementary Fig. 1 respectively). The PASp domain alone in both the citrate bound and the citrate free states has been studied by crystallography and solution NMR[6]. However, structural changes in PASc which would reveal the transmembrane signaling mechanism remain uncharacterized. Previously, we reported a piston-like motion in the PASp domain along with dynamics changes in the PASc domain of CitA from the thermophilic bacterium *Geobacillus thermodenitrificans* (GT) upon switching from the free to the bound state by citrate binding[6]. PAS-containing crystal structures suggest a variability of structural arrangement between the PAS core and its N-terminal helix[16], including the observation of both a parallel and a 120° rotated dimer in one of the cytosolic PAS domains in the HK KinA[17]. In the context of this paper, PAS core refers to the PAS domain without the N-terminal helix. In addition, mutagenesis studies have also provided evidence for dimer rearrangement[16] upon the signaling event.

In this work, we report structural changes in the TM2 helix and its associated linkers at single-residue resolution, by using chemical shift information from [1]H-detected solid-state NMR spectra. We used a bilayer-embedded CitA construct, referred to as the CitApc construct, that contains all essential elements for transmembrane signaling (PASp, TM helices and PASc) and the fully functional R93A mutant[6]. The R93A mutant is used to access the citrate free state, since the wild type cannot be produced without citrate due to very tight binding. With this construct, we detected a nanometer scale dimer rearrangement of the cytosolic PAS domains that occurs with the addition of citrate.

## Results and discussion
### P-helix formation and TM2/PASc-linker rearrangement
We recorded [1]H-detected MAS NMR spectra of reconstituted CitApc in DMPC/DMPA bilayers in order to obtain chemical shift assignments that are sensitive to protein secondary structure. The extracytoplasmic helix (P-helix) at the C-terminal end of the receiver domain (PASp) is a consensus structural feature of HKs[18] and has been confirmed previously with [13]C-detected solid-state NMR for the residue L154 in the ligand-bound state[6]. Based on chemical shift assignment (Fig. 1A, B, BMRB ID: 51759 and 51760), we are able to extend this helical characterization to four more amino acids up to I158, which is only one residue away from the predicted membrane embedded region based on cysteine accessibility in the homologous protein DcuS[8] (Fig. 1C). This result is in agreement with the conversion of the disordered region downstream of the last β-strand in the PASp core (residues 152-158) in the citrate-free state to a helix (P-helix) extending to the TM2 helix, in the bound state (Fig. 1D). This helix formation supports the proposed piston-like upward motion of the TM2 helix upon citrate binding[6]. We note that the anionic residues 156 and 157 might be modulated by the membrane composition[19–21], which is far more complex in native membranes. Here, our focus is on the structural response of CitApc to citrate. The relevance of the use of a simplified lipid bilayer is justified by the response of full-length CitA to citrate binding as observed via [31]P NMR tracking of ATP hydrolysis (Supplementary Fig. 2) when reconstituted in DMPC liposomes.

Chemical shift assignments of the linker between the TM2 and PASc[22] in CitApc also indicate changes in secondary structure upon addition of citrate (example assignment spectra for this region are shown in Supplementary Figs. 3–5). Details of the assignment process based on a series of 3D spectra are included in the methods section.

With the upward piston-like motion of the TM2 helix, 180GAVG183 at the N-terminus of the TM2/PASc-linker undergoes a transition from mostly extended (β-sheet-like) secondary chemical shifts to α-helical secondary chemical shifts in the bound state. Assignments in this region are confirmed by solvent transfer experiments, with lipid contact found for transmembrane residues, but absent for characteristic and well-separated resonances of the PAS domains[23] (Supplementary Figs. 6 and 7). Since both glycine and alanine have a higher tendency to form an α-helix in a hydrophobic environment[24], the upward shift of the 180GAVG183 region into the membrane facilitates its contraction into a helical conformation. At the C-terminus of the TM2/PASc-linker, the 192KK193 motif does the opposite and loosens into a more extended structure when citrate is bound. This is seen from a change in secondary chemical shift from positive to negative for residues K192 and K193 (Fig. 1E) and visualized in a structural model calculated based on torsion angle restraints predicted by TALOS-N[25] (Fig. 1F). Cysteine cross-linking performed in DcuS TM2 and the linker also showed a reduced dimerization of the linker region in the presence of the ligand[9], which is compatible with a more extended conformation of the linker. Furthermore, a loosened 192KK193 linker would explain the mobility in the PASc domain in the citrate-bound state indicated by the fewer peaks above the detection limit of the PASc domain in cross-polarization (CP) based solid-state NMR spectra of CitApc with both [13]C and [1]H detection[6] (Fig. 1A, B). This internal dynamics of the PASc domain limits the amount of assignable residues for unambiguous structure calculation of the PASc domain in CitApc by NMR (Fig. 1A, B). Therefore, we applied [19]F NMR-based distance measurements guided by our crystal structures of the PASc domain[26] (*vide infra*).

### PASc domain quaternary structural changes in solution
We observed more than one quaternary structure for the soluble PASc dimer both in crystals and in solution, in line with biochemical evidence suggesting that modulation of the PASc dimer interface is involved in the signaling process[27–29]. We first generated functional mutants in the PASc domain from *G. thermodenitrificans* CitA, based on functional mutation loci in *E. coli* DcuS[26] identified through systematic mutagenesis and sequence alignment between *E. coli* DcuS and *G. thermodenitrificans* CitA. The N288D CitA was derived from N304D of DcuS which kept DcuS in the OFF state[29]. When the function of the N288D CitA mutant was assessed by a β-galactosidase assay performed in *E. coli* CitA reporter strains (Fig. 2F), we found that N288D CitA unlike N304D of DcuS was fully functional, *i.e.*, activated by citrate. The N288D mutation in the PASc domain retains the same function as the wild-type (WT) CitA. And, interestingly, the isolated N288D PASc domain shows an anti-parallel dimer arrangement in the crystal structure (Fig. 2A, PDB code: 8BJP), whose N-terminal helices run in opposite directions and, together with the N-terminus of the major loop of the PAS domain core (N245 to T252), form an anti-parallel dimer interface. In contrast, in the crystal structure of the WT PASc domain[26] (Fig. 2B), the N-terminal helices of the dimer run in the same direction and form a parallel dimer interface. Notably, anti-parallel and parallel dimer forms are not domain-swapped dimers as seen in the cases of human βB2-crystallin[30] and cyanovirin-N protein[31], since they share the same monomer structure. The parallel dimer arrangement was previously reported in VicK[11] but the anti-parallel arrangement had not been reported. Both the WT PASc and N288D PASc eluted as a dimer from a size exclusion column (SEC) down to a concentration of 1 μM, maintaining the dimer state in solution as hypothesized for membrane bound CitA (Supplementary Fig. 8). In agreement, the parallel or anti-parallel dimer arrangements were found to be the dominant forms in solution, respectively, for the WT or N288D PASc domain based on the following two methods. First, the surface accessibility profile was measured from solvent paramagnetic relaxation enhancement (sPRE) and the measured intensity ratio profiles of the N-terminal helix distinguish between the dimer forms of WT

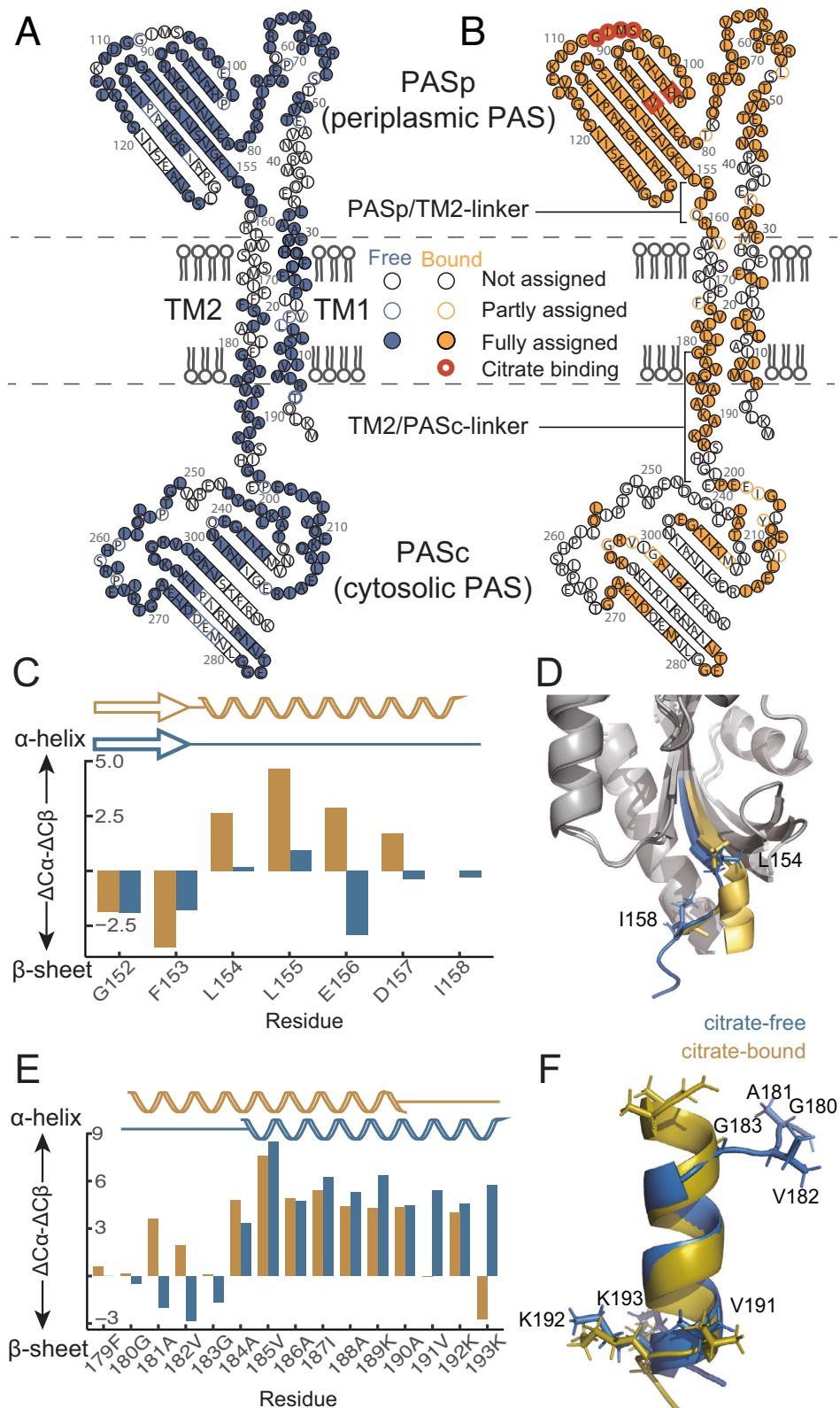

**Fig. 1 | Sequence specific assignment with ¹H-detected MAS NMR revealed secondary structural changes of CitA upon citrate binding. A** Assigned residues shown in the topology map in the citrate free state and **B** the citrate bound state of the CitApc construct. Citrate binding residues are indicated by red circles in (**B**). Chemical shift assignments are as recorded in Supplementary Date Files 1 and 2. **C** Citrate binding causes chemical shift changes in the PASp/TM2-linker. The contraction of the overall β-scaffold of PASp leads to a random coil to α-helix transition in the PASp/TM2-Linker. **D** Superposition of the citrate-bound Gt PASp domain structure (PDB: 8BGB, [https://doi.org/10.2210/pdb8BGB/pdb]) with the citrate-free structure (PDB: 8BIY, [https://doi.org/10.2210/pdb8BIY/pdb], Supplementary tables 6, 7 and 8). The border of this change ranges from L154 (previously described in Salvi *et al.*[6].) to I158. **E** Secondary chemical shift changes in the TM2/PASc-linker. **F** The structural model calculated using torsion angle restraints predicted with TALOS-N visualizes the N-terminal contraction and C-terminal loosening of TM2/PASc-linker upon citrate binding.

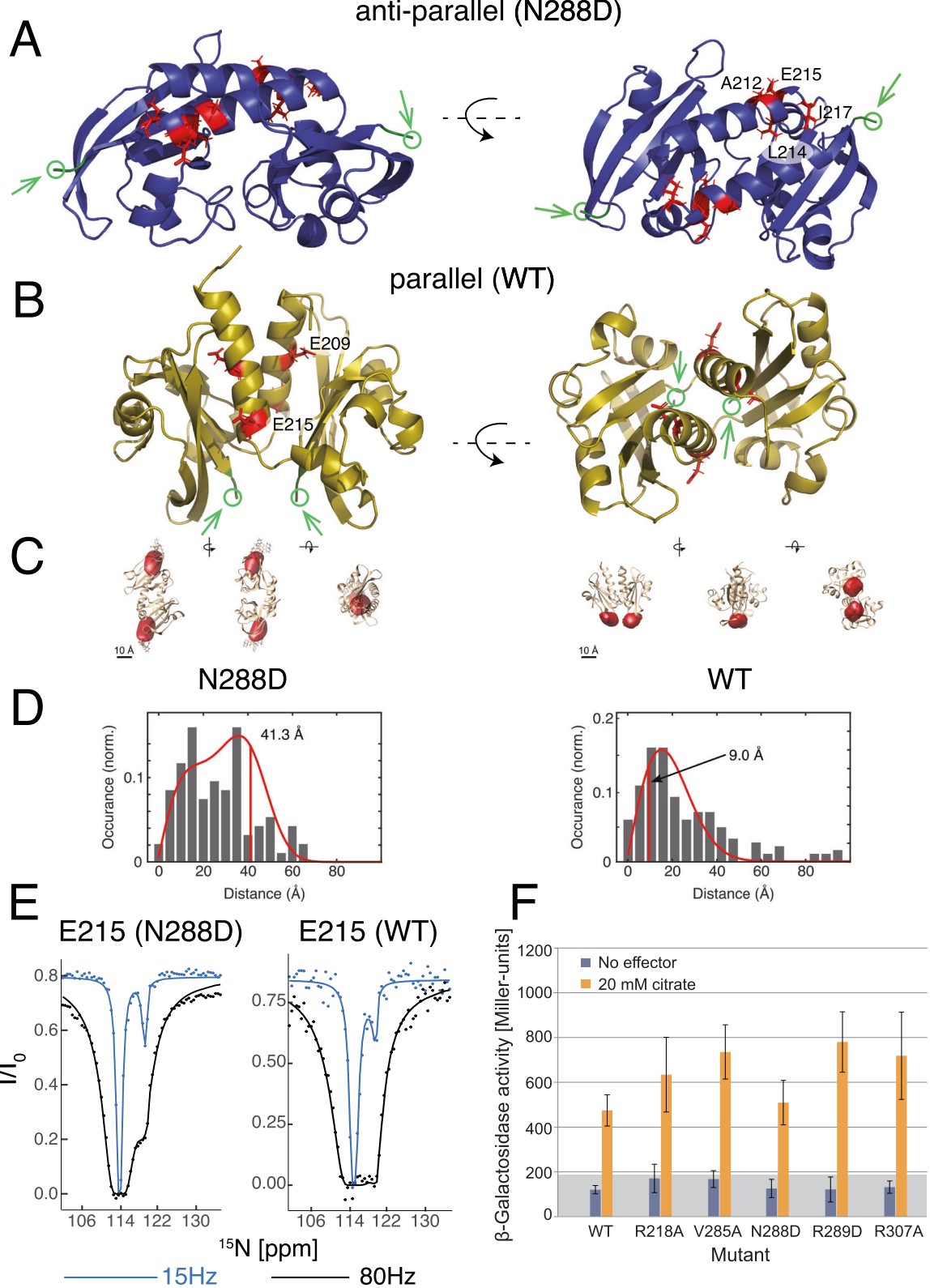

(BMRB ID: 51764) and the N288D mutant (BMRB ID: 51765) PASc (Supplementary Fig. 9). Second, cryogenic optical localization in 3D (COLD) was used to measure the distance between two fluorophores covalently attached to the protein[26]. The isolated PASc domains were labeled at the C-terminal residue 308 via introduction of an N to C mutation and attachment of the dye Atto647N. An inter-N308C distance of 41.3 Å was obtained in the N288D mutant PASc, which agrees with the expected distance in the anti-parallel dimer arrangement

(Fig. 2C, D). While some populations of dimers at smaller separations in Fig. 2D cannot be excluded, the distribution of large distances about 41.3 Å indicates that the antiparallel arrangement dominates. Regarding WT PASc, an inter-N308C distance of 9.0 Å is observed, which fits the expected distance based on the crystal structure of the parallel dimer[26]. This finding has recently been corroborated by MINFLUX[32].

Transition between conformations of the isolated PASc domain is observed by chemical exchange saturation transfer (CEST), which

**Fig. 2 | Residues at the dimer interface of isolated PASc show CEST exchange profiles, and COLD distance measurement confirms dimer forms in crystals.** Residues showing CEST exchange (red) plotted on the crystal structure of (**A**) the N288D PASc mutant (PDB: 8BJP), which has the anti-parallel dimer arrangement and (**B**) the WT PASc (PDB: 5FQ1), which has the parallel dimer arrangement. The C-terminal residue N308 is shown in green. **C** Reconstructed volumes (red) of the fluorophores plotted at an isovalue of 0.68 and overlayed with the crystal structure of the N288D mutant (left) and WT (right) PASc. Views from three orthogonal directions are shown. **D** Histogram of the measured projected distances and the expected distribution (red curve) as determined by fitting a model function, which is the convolution of a projection function and the Euclidian norm of a bivariate normal distribution of the two determined fluorophores positions. The red vertical lines at 41.3 Å (N288D mutant, left) and 9.0 Å (WT, right) show the resulting expectation value of the inter-fluorophore distance. **E** Example CEST profile and fitting of the residue E215. The fractional populations of the minor state in the WT and N288D mutant PASc domain are 3.8% and 3.3%, respectively, based on the fitted exchange rates (Supplementary Tables 9 and 10). **F** The N288D mutant CitA has the same activity and reactivity to citrate as the WT CitA according to a β-galactosidase assay performed in *E. coli* reporter strain. The number of replicates, N, was 12 for each experiment, with the following Miller Units MU for the center lines and the lower/upper limits: mean ± SD: WT (wild type) 476 (±69) MU, variant R218A 632 (±166) MU, variant V285A 734 (±121) MU, variant N288D 506 (±100) MU, variant R289D 786 ± 141) MU, and variant R307A 715 (±197) MU.

detects lowly populated states when they undergo millisecond (ms) timescale conformational exchange with the main population[33]. For the PASc domain in CitA, this CEST effect was observed mainly in the residues from the N-terminal helices that form the dimer interface in both the anti-parallel and parallel arrangements (Fig. 2E and Supplementary Fig. 10). This indicates a dimer interface rearrangement of the PASc domain in isolation where the PASc domains switch most probably between the parallel and anti-parallel dimer forms, even when CEST does not reveal the identity of the conformations exchanging. Although this dimer exchange seen in the WT PASc is functionally more relevant, its occurrence in the N288D PASc mutant helps to rationalize why this point mutation does not disturb the functionality of the full length CitA (Fig. 2F). MINFLUX[32] of the PASc domain confirms the conformation switching in WT. Still, connecting a PASc domain conformation with a specific citrate receptor state requires a membrane embedded construct.

### Larger inter-dimer distance change at the PASc C-terminus

For this purpose, we measured the characteristic dimer distance at residue 308 in the CitApc construct embedded in lipid bilayers (Fig. 3A). The same mutation used previously in COLD measurements was introduced for site directed $^{19}$F labeling, and the method Center-band Only Detection of EXchange (CODEX)[34] combined with dynamic nuclear polarization (DNP) was applied. DNP was important to provide sufficient sensitivity to detect the fluorine signal from the singly $CH_2COCF_3$-tagged protein in the liposome-embedded sample. Equally crucial is the combination of sensitivity enhancement provided by the three magnetically equivalent fluorine atoms in the $CF_3$ group, and proton to fluorine cross polarization (CP) (Supplementary Fig. 11). An inter-$CF_3$ group distance shorter than about 20 Å is measurable by fitting the CODEX decay curve, and a distance larger than 20 Å can be concluded if there is an absence of decay within the 250 ms of CODEX. The CODEX experiment performed in both the free and bound state of the CitApc construct can then distinguish between the anti-parallel and parallel dimer in the PASc domain, where the former has an inter-N308C distance of 40 Å and the latter of 11 Å based on their crystal structures (Fig. 3B, C) and the COLD measurement (Fig. 2D, F). A decay in the CODEX signal to half of the reference signal was observed when citrate was bound (Fig. 3A). Fitting of the decay curve resulted in a distance of 13.6 ± 3 Å, corresponding to the inter-N308C dimer distance expected in the parallel dimer within experimental error and tag flexibility. It is difficult to completely remove citrate, such that our citrate free sample still contains about 30% population of the bound state, as estimated from the intensity of the H96 side chain peak in the (H)NH spectrum that is characteristic of the bound form (Supplementary Fig. 12C). Substantially less CODEX decay was observed in the citrate free sample (Fig. 3A), indicating a larger inter-N308C dimer distance in the citrate free state as compared with the citrate bound state.

Considering the 30% citrate bound population, a population weighted fit reveals a distance beyond 20 Å for the citrate free state, revealing a substantial decrease in the inter-N308 dimer distance upon

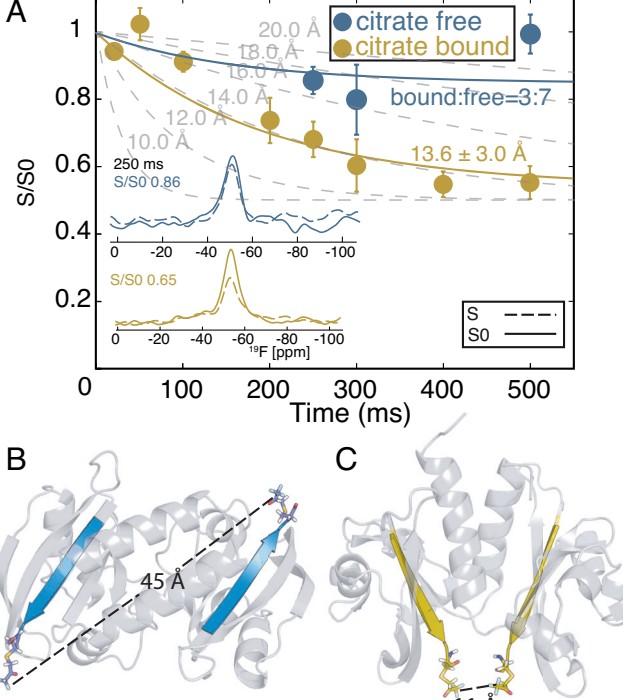

**Fig. 3 | The C-terminal dimer distance of $CH_2COCF_3$ PASc observed by CODEX dephasing for both bound (yellow) and free (blue) states of CitApc.** At the mixing time of 250 ms, the bound state CODEX signal decays to 0.65 of the reference experiment (**A**, inset yellow); the decay rate fits to an inter-$CF_3$ distance of 13.6 ± 3.0 Å, with the exponential decay fit curve of $0.5*e^{-0.0053*t} - 0.5$, matching the expected inter-dimer distance at the C-terminus of the parallel dimer (**C**). In contrast, the free state CODEX signal is almost the same as the reference experiment (**A**, inset blue) through all mixing times, corresponding to a large inter dimer distance at the C-terminus of more than 20 Å, agreeing with the anti-parallel dimer (**B**). Example CODEX decay curves at different inter fluorine distances are shown (**A**, gray). 30% of the bound state CitApc protein present in the citrate free sample caused the minor CODEX decay. Error (taken at 1 SD), signal and noise levels of each individual measurements are recorded in Supplementary Tables 11 and 12. The CODEX decay curve could be acquired beyond the $^{19}$F $T_1$ of 321 ms (Supplementary Fig. 14), thanks to an eight-fold DNP signal enhancement (Supplementary Fig. 11B).

citrate binding. The nanometer-scale change in the inter-dimer distance at the C-terminus of the PASc domain as identified with CODEX confirms a rearrangement in quaternary structure that is compatible with an anti-parallel to parallel dimer transition upon ligand binding. Importantly, $CH_2COCF_3$-tagged CitA is fully functional and responsive to citrate binding based on the ATP hydrolysis rate tracked using $^{31}$P NMR[35] (Supplementary Fig. 2). Upon citrate binding, a threefold increase in CitA's ATP hydrolysis rate was observed, similar to the level of activity increase found using an in vivo assay (Fig. 2D).

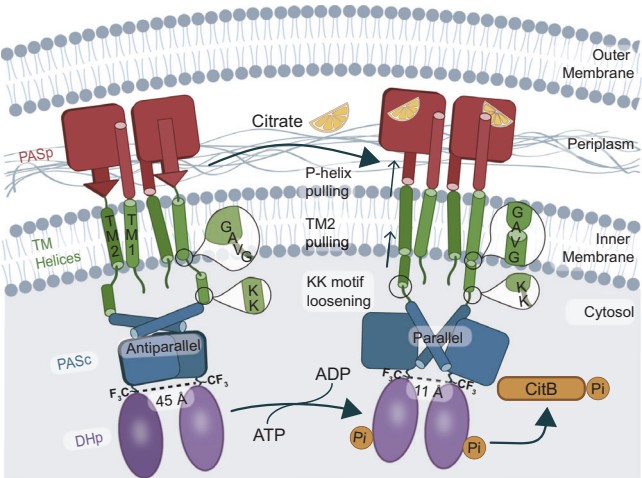

**Fig. 4 | Mechanism of CitA activation by citrate, derived from a combination of structural studies of PASp, PASc and CitApc constructs and activity assays of full-length CitA.** Important structural changes underlying the transmembrane signaling process of CitA include the P-helix formation, piston-like shift of the TM2 helix into the periplasm, an extension in the $_{192}$KK$_{193}$ motif, and the smaller inter-dimer distance at the PASc C-terminus. Altogether, the citrate binding has caused a large intracellular structural change, which is available to potentiate a difference in the CitA cross-phosphorylation and consequently, phosphorylation and activation of the RR CitB.

While we investigated CitA from GT, there is some evidence that PASc domains of other sensory kinases switch dimer arrangements as well: Cys-scanning mutagenesis targeting the β-sheet of the cytoplasmic PAS domain of sensor kinase BvGs, resulted in changing patterns of S-S bond-mediated cross-linking dependent on the signaling state, in line with changing quaternary structure of the PAS domain dimer during signal transduction[27]. Replacing the oxygen-sensing PAS domain of the B. japonicum FixL kinase with a photosensory PAS domain[36] resulted in a chimeric protein retaining the catalytic efficiency of FixL but responding to blue light instead of oxygen. This led to the conclusion that a high degree of similarity in the mechanisms of the parent chemosensor and the chimeric photosensor is implied[36].

Our structural findings related to transmembrane signaling of the citrate binding sensor kinase CitA are illustrated in Fig. 4. The large change in quaternary structure upon citrate binding detected by CODEX, is represented by the high-resolution structures of PASp and PASc dimers, and together with changes in secondary structure identified by NMR chemical shifts, the mechanism of transmembrane signaling is revealed. Structural changes include the P-helix formation, which promotes the piston-like pulling of the TM2 helix, and the amplification of this upward motion by the helical conversion of residues $^{180}$GAVG$^{183}$, which border the membrane and the cytosol. This contraction into a helical conformation can be explained by the change of solvent environment as these residues are pulled up from aqueous phase into the membrane, becoming part of TM2. This potentially induces a hydrophobic mismatch of the TM2 in one of the states unless this is relieved by a change in the helix tilt. This contraction of the TM2/PASc-linker is compensated only partly by loosening of the $^{192}$KK$^{193}$ motif close to the N-terminus of PASc, leading to a change in the residue 308 dimer distance from more than 20 Å to 13.6 ± 3.0 Å based on CODEX data. These distances are consistent with a transition from the anti-parallel to a parallel conformation captured in crystal structures of the isolated PASc domains, in which the N308 dimer distance changes from 45 Å to 11 Å. Substantial dimer reorganization is not without precedent, as a 120° rearrangement has been observed in an isolated PAS-A domain[17], though different from the dimer reorganization observed in the case of CitA PASc domain. A

transition to a parallel dimer, which aligns the N-terminal helices in the citrate bound state is likely caused by the shortening of the TM2/PASc-linker. Mechanistically, these data suggest that the kinase core domains are held apart due to the 45 Å distance separating the C-termini of PASc in the citrate-free state, while they are brought together to turn on cross-phosphorylation in the citrate-bound state, based on an 11 Å distance between the C-termini of PASc in the citrate-bound state.

Several mechanistic details of CitA activation are similar to those observed in the HK NarQ, which has HAMP domains rather than PAS domains as both its receptor and cytosolic connector domains[37]. The crystal structure of NarQ without its kinase core revealed a piston-like motion for the TM helices upon activation similar to that detected in the PASp/TM2-linker and the TM2 helix of CitA. In contrast, the lever-like rearrangement of the NarQ HAMP domain is distinct from the structural rearrangement observed here in the PASc domain upon activation[37].

In summary, we have formulated a transmembrane signaling mechanism of CitA from the citrate binding at the PASp domain to the structural and dynamic changes in the PASc domain. Future studies will have to elucidate the arrangement of the PASc dimer in the context of the full-length CitA, and how its conformational dynamics affect the structure of the DHp dimer. Given the fact that a large number of HKs contain a cytosolic PAS domain, and PASc dimer switching would explain observations on other sensory kinases, our findings can be applied across a wide spectrum of bacterial HKs and also assist with the formulation of a uniform HK activation mechanism.

## Methods

### Cloning procedures

Most constructs for Gt CitA fragments used in this study have been described before[6]. Full length Gt CitA was cloned into vector pET16b. The mutations C12A, R93A and N308C in this full length Gt CitA expression construct as well as the N288D mutation in the pET28a-PASc domain construct[26] were introduced by PCR using a Quik-ChangeII Site-Directed Mutagenesis Kit (Agilent Technologies, see Supplementary Table 1 for the cloning and mutagenesis primers).

### Protein sample preparation

For producing approximately 100% deuterated, $^{15}$N, $^{13}$C-labeled C12A/R93A CitA1-309, CitApc, the plasmid coding for C12A/R93A CitApc was transformed into the strain C43 (DE3) (Imaxio, France). The R93A mutant was previously used in the isolated PASp domain to investigate its structure in both the citrate bound and citrate free form[6]. Without the R93A mutation, the PASp domain in Gt CitA binds the citrate molecule too tightly for a citrate free state to be produced. A single colony was transferred into a 2 ml minimal medium shaking culture and step-wise adapted to 100% D$_2$O, followed by expression in minimal medium with 100% D$_2$O, $^{15}$N-NH$_4$Cl as nitrogen source and $^{13}$C$_6$-D-glucose as carbon source. For additional reverse labeling with isoleucine, valine, phenylalanine and leucine, 150 mg/L L-isoleucine-d$_{10}$, 100 mg/L L-valine-d$_8$, 50 mg/L L-phenylalanine-d$_8$ and 150 mg/L L-leucine-d$_{10}$ (Merck) were added to the expression culture shortly before induction. Purification was performed according to the established protocol[6]. Briefly, after induction with IPTG, cells were incubated over night at 20 °C in a shaking culture before being harvested. The cell pellets were resuspended in TKMD buffer (50 mM Tris·HCl pH 7.0, 200 mM KCl, 5 mM MgCl$_2$, 5 mM β-mercaptoethanol, one spatula tip of DNAseI, 0.5 mM PMSF). The resuspended cells were lysed in a French pressure cell (20,000 psi). Cell debris was pelleted by centrifugation and supernatant spun down in an ultracentrifuge. The resulting pellet was re-suspended in Ni-NTA buffer (20 mM Tris·HCl pH 7.9, 500 mM NaCl, 10 mM imidazole, 5 mM β-mercaptoethanol, 0.5 mM PMSF, 4% Triton X-100 (v/v)) and diluted using Ni-NTA-buffer without Triton

X-100 to set the final detergent concentration to 0.8% (v/v). The protein was purified using a 5 mL Ni-NTA column (GE Life Sciences). After washing, the detergent was changed on-column by washing with Ni-NTA-buffer supplemented with 1% (v/v) laurydimethylamine-oxide (LDAO) instead of Triton X-100. For elution, the imidazole concentration was increased to 500 mM. Size exclusion chromatography (SEC) on Superdex 200 26/60 columns (GE Life Sciences) was performed as the final purification step using 20 mM Tris·HCl pH 7.4, 150 mM NaCl, 1 mM DTT, 0.3% LDAO as running buffer. The purified C12A/R93A CitApc protein was reconstituted into 1,2-dimyristoyl-sn-glycero-3-phosphocholine (DMPC) and 1,2-dimyristoyl-sn-glycero-3-phosphatic acid (DMPA) liposomes (with DMPC to DMPA molar ratio of 9:1) at a lipid:protein ratio of 75:1 (mol/mol). The liposomes were taken up in 20 mM sodium phosphate, pH 6.5, 5 mM sodium citrate. The citrate free sample is made by incubating the citrate bound CitApc sample with the citrate free buffer (20 mM sodium phosphate pH 6.5) at 40 °C for a week. This process is monitored by tracking the intensity of the H96 side chain with 2D (H) NH spectra (Supplementary Fig. 12). The liposome samples were packed in a Bruker 1.3 mm rotor by ultracentrifugation.

Full length C12A/R93A/N308C Gt CitA was expressed and purified following the established protocol for the C12A/R93A/N308C Gt CitApc construct with modifications. After purification by gel-filtration, the protein was loaded onto 4 ml Ni²⁺-NTA resin and eluted with 20 mM Tris/HCl, pH 7.4, 150 mM NaCl, 500 mM Imidazole, 0.694% (w/v) decylmaltoside (DM) and 0.5 mM Tris-(2-carboxyethyl)-phosphine hydrochloride (TCEP). After adding 6 mM ATP, the protein was reconstituted into DMPC liposomes at a lipid to protein ratio of 20/1 (w/w), removing the detergent with BioBeads (BioRad). The final concentration of the liposome-reconstituted protein in the buffer was adjusted to 100 µM. The choice of the lipid was determined by highest spectral quality and functionality of CitA.

The introduction of fluorine labels using cysteine alkylation by 3-bromo-1,1,1-trifluoroacetone (BTFA) was done according to the protocols published by the group of Schofield[38–40]. Tagging of the C12A/R93A/N308C-Gt CitA protein with BTFA was performed using 50 mM sodium phosphate, pH 7.0, 200 mM NaCl, 0.3% (w/v) LDAO in the gel-filtration step. Subsequently BTFA was added drop-wise to the protein solution as described for the BTFA-tagged CitApc, from a 100 mM stock solution in 50 mM sodium phosphate, pH 7.0, 200 mM NaCl. After incubation overnight on ice, the protein was loaded onto 4 ml Ni²⁺-NTA resin to switch to 0.694% DM and reconstituted in DMPC liposomes at a lipid to protein ratio of 20/1 (w/w), with 6 mM ATP added to the buffer. The final protein concentration was adjusted to 100 µM. To obtain the citrate bound state, buffer containing 5 mM sodium citrate was added to the sample and equilibrated overnight at 4 °C.

## sPRE measurement
Solvent paramagnetic relaxation enhancement (sPRE) was measured and assessed according to published protocols[41,42]. ¹⁵N WT and N288D PASc were doped with 2.5 mM of Gd-HP-DO3A, for measurement of the paramagnetic ¹⁵N-HSQC spectra. The intensity ratio between the paramagnetic and the diamagnetic (without Gd-HP-DO3A) ¹⁵N-HSQC spectra ($I_{sPRE}/I_0$) was used as the experimental sPRE value. A lower value is indication of higher solvent accessibility due to paramagnetic relaxation from the soluble gadolinium relaxation agent. All the spectra were recorded on a Bruker 400 MHz spectrometer equipped with a 5 mm triple channel room-temperature probe using an inter-scan delay of 2.5 s.

The predicted sPRE values from the crystal structures were obtained from the algorithm described by Öster et al.[42] The output of the algorithm is a number indicating the linear enhancement of the relaxation rate of a spin as a function of the paramagnetic agent concentration and high value corresponds to higher solvent exposure.

## CEST experiment
¹⁵N CEST data were recorded as described by Vallurupalli et al.[33] ¹⁵N radio-frequency field strengths of 20 and 80 Hz were applied together with ¹H decoupling of 3.5 kHz during a relaxation delay of 400 ms. CEST data consisted of a series of 2D spectra, acquired in an inter-leaved fashion, corresponding to ¹⁵N irradiation offsets incremented in steps of 0.5 ppm. Additionally, a reference experiment for which the relaxation delay is set to 0 was recorded. CEST fitting was done using an in-house software described previously by Carneiro et al.[43].

## ³¹P NMR ATP hydrolysis assay
Before the ³¹P NMR activity assay, approximately 150 µL full length CitA liposome sample was pelleted and then resuspended in 20 mM Tris/HCl pH 6.5, 50 mM KCl, 10 mM MgCl₂, 0.5 mM EGTA and 6 mM ATP, 0.5 mM Pefabloc (Roth, Germany), 0.02% NaN₃. After the measurement was finished, the samples were centrifuged and the pellet was weighed to compare the amount of protein in the citrate bound and the citrate free sample. The final reaction rate normalized by the weight was compared.

The detection of ATP in solution, along with its degradation product, the inorganic phosphate, with NMR is straightforward, owing to the 100% natural abundance of the NMR-active nucleus ³¹P and the large chemical shift separation[44–46] of the free phosphate. 1D phosphorus spectra were measured with a Bruker 400 MHz magnet. The spectra for full length un-tagged R93A/C12A/N308C CitA were measured with a room temperature QXI H/P-C/N/D five channel probe with a B₁ field corresponding to a 7.5 kHz nutation frequency. The spectra for CF₃-tagged R93A/C12AN/308 C CitA were measured with a room temperature TBO H/F-C/N/P-D probe. All spectra were measured with an inter-scan delay of 1 s and an offset at -15 ppm. The number of scans was set to 1920. 10% D₂O was used in all samples as lock signal. The measurement temperature for all CitA full length constructs was 37 °C. The initial speed of the reaction was used to measure the kinase activity since the only reactant (ATP) is added in large excess to the enzyme. ATP is stable under the experimental condition (Supplementary Fig. 13), so its auto-hydrolysis does not need to be taken into account. The error is estimated from the noise level of the spectra. According to pseudo-zero-order kinetics (high excess of substrate), the change in reactant concentration is approximately linear. This appliesat very short reaction time, when the free phosphate (Pi) production approaches a linear equation with the slope corresponding to the ATP hydrolysis rate.

## β-Galactosidase assay
β-Galactosidase as a reporter of the activity of CitA of *G. thermodenitrificans* was tested in the *E. coli* strain, IMW549 (*citA⁻* and *citC–lacZ*), that is deficient of *E. coli citA* but carries an *E. coli citC-lacZ* β-galactosidase reporter fusion[12]. The strain was transformed with plasmids encoding CitA of *G. thermodenitrificans* or variants R218A, V285A, N288D, R289D, or R307A thereof, (pMW1652 and variants) together with CitB *of E. coli* (pMW1653). The bacteria were grown without or with citrate (20 mM), and the *citC–lacZ* reporter gene activity (mean ± SD) was measured[12] in three biological and four technical replicates each (in total $N = 12$).

## Sequence specific assignment and statistics
We were able to acquire and assign high frequency MAS NMR spectra in both the citrate bound and citrate free states, of homogeneous samples that were fully deuterated (Supplementary Fig. 3). The sequence specific assignment process of the linker from residue A181 to K193 in citrate free and citrate bound state using the 3D spectra is shown in Supplementary Fig. 4. To further confirm the assignment in the TM helices, (which have a higher abundance of the residue types of I, V, F and L), we measured samples in which these residues were reverse labeled (without ¹³C and ¹⁵N labeling). Resonances belonging to

I, V, F and L are expected to have no intensity in the spectra (Supplementary Fig. 5) from this reverse labeled sample, while in practice, incomplete elimination of valine signals was observed.

$^1$H-detected solid-state NMR spectra of CitApc, except the (H)N(CA)(CO)NH spectrum in the bound state, were measured at the field of 850 MHz with a Bruker 1.3 mm H/C/D/N four channel probe. The (H)N(CA)(CO)NH spectrum in the bound state was acquired at the field of 800 MHz with a Bruker 1.3 mm H/X/Y three channel probe. The peak lists extracted from the assignment experiments were used in automated assignment by FLYA[47], which aided in the initial process of sequence specific assignment. The lipid transfer H(H)NH spectra were not used for this purpose. Spectra were processed with Topspin and assigned using Collaborative Computing for NMR (CCPN) analysis[48].

The sample temperature was set to 293 K adjusted with the chemical shift of water, using a cooling gas flow of 1350 L/h at 243 K. Chemical shifts of $^{13}$C are referenced relative to sodium trimethylsilylpropanesulfonate (DSS). The following 3D spectra were acquired[49–51]: (H)CANH, (H)CA(CO)NH, (H)(CA)CB(CA)NH, (H)(CA)CB(CA)(CO)NH, (H)CONH, (H)(CA)CO(CA)NH, (H)N(CO)(CA)NH. Additionally, a 3D lipid transferred experiment H(H)NH[23] is acquired to confirm the assignment in the membrane contacting regions. Details of the experimental parameters are summarized in Supplementary Tables 2, 3, 4 and 5.

## X-ray crystallography

All constructs were expressed and purified as described before for the WT PASc[6,26]. PASp wildtype and mutant protein were expressed in *E. coli* strain BL21(DE3), the N288D mutant PASc protein was expressed in selenomethionine-labeled form in methionine-auxotrophic *E. coli* strain B834 (DE3) (Novagen). All three fragments were purified by immobilized metal affinity chromatography on Ni-NTA resin (Qiagen). The N-terminal His-tag was removed with Tobacco Etch Virus protease and final purification by size exclusion chromatography was performed on a Superdex 75 16/60 column (GE Healthcare). The purified WT and R93A mutant PASp proteins were dialyzed against 20 mM HEPES, pH 7.0, 50 mM NaCl. Protein concentration was adjusted to 20 mg/ml using a Vivascience 10 kDa MWCO concentrator. Finally, 1 mM sodium citrate was added to the WT Gt PASp sample. The N288D PASc protein was dialyzed against 20 mM sodium phosphate pH 6.5, 50 mM NaCl and the protein concentration was adjusted to 35 mg/ml.

Crystals of the three proteins were obtained by the vapor-diffusion technique with sitting drops. 100 nl of protein solution were mixed with 100 nl of well solution (1 M tri-sodium citrate, 0.1 M imidazole, pH 9.0 for WT PASp and 0.2 M lithium sulfate, 0.9 M sodium dihydrogen phosphate, 0.6 M di-potassium hydrogen phosphate, 0.1M N-cyclohexyl-3-aminopropanesulfonic acid (CAPS), pH 10.6 for the R93A mutant PASp; 0.4 M magnesium chloride, 0.1 M Tris/HCl, pH 8.5, 20.5% PEG 8000 for the N288D PASc mutant). Crystals grew within one week and were cryoprotected by transferring them for one minute to well solution supplemented with 30 % glycerol (WT PASp), 33 % glycerol (R93A PASp) and 20% glycerol (N288D PASc) and flash-cooled by plunging them into liquid nitrogen. Diffraction data sets were collected at beamline X10SA, SLS, Switzerland (PILATUS 6 M detector, Dectris). For PASp and its R93A mutant, only native data were measured at 100 K at 1.0 Å (see Supplementary Tables 6 and 7). Three data sets (peak, inflection and high-energy remote, see Supplementary Table 8) were collected at 100 K for the N288D PASc. All data were processed with XDS[52] and scaled with SADABS (Bruker AXS). Space group determination and statistical analysis were done with XPREP (Bruker AXS).

The structure of the wildtype PASp domain was solved by molecular replacement with PHASER[53] using the crystal structure of the citrate bound *Klebsiella pneumoniae* PASp domain[5] (PDB code: 2J80) as search model. Refinement was performed initially with Refmac[54] alternating with manual model building in Coot[55]. Final refinement was

performed with phenix.refine[56]. In the final model 97.84% were in the favored region, 2.16% were in the allowed region of the Ramachandran plot. The structure of the R93A mutant was also solved by molecular replacement with PHASER[53] using the crystal structure of the WT Gt CitA PASp domain as search model. Refinement was performed as described for the WT domain. In the final model 97.02% were in the favored region, 2.73% were in the allowed region of the Ramachandran plot, 0.25% were outliers. Refinement statistics for the WT and mutant Gt PASp domain structures are shown in Supplementary Tables 6 and 7. The structure of the N288D PASc domain was solved by multi-wavelength anomalous diffraction. Normalized difference structure factors were calculated using SHELXC, followed by solution of the substructure with SHELXD. Phasing statistics are shown in Supplementary Table 8. Phase extension, density modification and main chain auto-tracing was done with SHELXE[57]. Ten cycles of automatic model building alternated with structure refinement by ARP/wARP[58] resulted in modeling of 98 % of the residues. Refinement was done by positional and B factor refinement in REFMAC5[54] alternating with manual model building in COOT[59]. In the final model 96.17% of the residues were in the favored region of the Ramachandran plot and 3.83% in the allowed region. Refinement statistics for the N288D PASc domain structure are shown in Supplementary Table 8.

## COLD measurement and statistics

The details of the COLD experimental setup, sample preparation and data analysis are described in SI Material and Methods published previously in Weisenburger et. al[26]. We recorded COLD measurements on a custom-built cryogenic microscope at liquid helium temperature ($T = 4.3$ K). Samples were prepared by spin-coating diluted PASc N288D proteins conjugated with Atto647N-maleimide (AttoTec GmbH) at N308C onto thoroughly cleaned UV-grade fused silica coverslips (ESCO). We estimated the protein concentration by UV/VIS spectroscopy to be about 15 µM using an extinction coefficient of $\epsilon_{280} = 38{,}400\ \mathrm{M^{-1}\,cm^{-1}}$. The labeling efficiency was determined to be about 65%. Raw images of the fluorescent recordings were analyzed using custom-written software in MATLAB (The MathWorks). Localization images were generated and then fed into a single-particle reconstruction using the subspace EM algorithm[60]. To compare our results with the protein structure, we attached Atto647N fluorophores to the N308C positions at the C-termini of the crystal structure (PDB ID: 8BJP) by modeling. We used the point symmetry of the volume maps of the 3D reconstruction and fitted them to the crystal structure using UCSF Chimera, which was also used for visualization. The resolution of the 3D reconstruction was estimated by Fourier shell correlation of two half data set reconstruction using Free FSC (IMAGIC). Comparison with the half-bit criterion yields a resolution of 1.9 Å. We have performed a total of 13 similar experiment sessions over a period of seven months. The published data include 2133 identified proteins, which after filtering resulted in 188 proteins at a yield of 8.8%. The filtering criterion was localization precisions better than 6.5 Å, as described in Supplementary Table 2 from Weisenburger *et. al.*[26] We graphically display distributions by histograms for all relevant data in Fig. 2C, D. To extract quantitative information, we plot the histogram of measured distances together with a fit based on a Monte Carlo model and simulated annealing, also considering the localization precision values. In a simple two-dimensional distance measurement scenario, one expects the measured values to distribute symmetrically about a most probable value. In our case, the distances represent the projection of distances in a randomly oriented 3D arrangement, leading to an asymmetric distribution due to a cosine dependence. The final asymmetric distribution stems from the convolution of a projection function and the Euclidian norm of a bivariate normal Rice distribution to take into account the statistical error of each projection. Because of this skewed distribution, the expectation value does not coincide with the peak position. For the 3D reconstruction and

orientation fitting, datasets above a threshold localization precision of 6.5 Å were excluded.

## CODEX measurement under DNP

All liposome samples are prepared for DNP experiments by adding TEMTriPol powder to the liposome sample until the solubility limit is reached (around 20 mM). All samples are packed into 2.5 mm Phoenix MAS rotors and shock frozen by plunging into liquid nitrogen.

All CODEX experiments were measured with a previously published pulse program[34,62] (Supplementary Fig. 11) on a Bruker 600 MHz magnet, with a 2.5 mm Phoenix DNP probe equipped with H/F double channel tuning at 20 kHz MAS and a set temperature of 90 K.

The initial $^1H$ to $^{19}F$ CP contact time was 1500 µs with 41.6 kHz $^{19}F$ irradiation and 51 kHz $^1H$ irradiation. The $^{19}F$ and $^1H$ hard pulse power was 65.8 kHz and 53.2 kHz, respectively. The $^1H$ TPPM decoupling power was set to 50 kHz. The inter-scan delays were optimized for each sample individually. The offset of $^{19}F$ and $^1H$ was −74 ppm and 3 ppm, respectively. The sweep width for all $^{19}F$-detected experiments was set to 400 ppm.

CODEX is an NMR method based on spin diffusion and CSA recoupling to investigate slow dynamics[34] and oligomer number[61,62]. The rate of spin diffusion is parameterized by a factor F(0). This empirically determined overlap integral (physically a calibration of the rate of spin diffusion at a particular MAS frequency and magnetic field) can then be used to fit distances in unknown systems. F(0) is the overlap of the single quantum lineshapes of the two spins[61]:

$$F_{ij}(0) = \int_{\infty}^{-\infty} f_i(\omega - \omega_i) \cdot f_j(\omega - \omega_j) d\omega \qquad (1)$$

$\omega_i$ is the center of each peak, and $f_i$ and $f_j$ describe the intensity of the signal of each spin. The rate of spin diffusion is also affected by the dipolar coupling, $\omega_{ij}$, between two spins such that the spin diffusion rate between $i$ and $j$ ($k_{ij}$) is:

$$k_{ij} = 0.5\pi \cdot \omega_{ij}^2 \cdot F_{ij}(0) \qquad (2)$$

The spin diffusion rate can be calculated from the CODEX curve using models described in literature[55,61,63], by the following equations:

$$M(t) = e^{-Kt}M(0) \qquad (3)$$

Equation 3 describes the magnetization evolution, $M(t)$, with spin diffusion time (t). $K$ is the $n$-dimensional exchange matrix containing the rate constants $k_{ij}$ between exchange of two spins $i$ and $j$ (see Eq. 2), where $n$ is the total number of orientationally different spins in close enough vicinity to undergo magnetization exchange.

The spectral overlap function $F_{ij}(0)$ generally has substantial variation from sample to sample. Here, we took the previously published $F_{ij}(0)$ value of 30 to 60 µs, determined at a magnetic field of 14.1 T, under 20–40 kHz MAS for a variety of fluorine containing molecules[64]. Nevertheless, since $\omega_{ij}$ depends on the inverse cube of the distance, changes in F(0) affect the determined distance according to the sixth root. This means that a two-fold range in F(0), e.g., 30–60 µs will only introduce about 12 percent error in the determined distances.

Monoexponential fit of the CODEX curve was performed using the GNUPLOT fit function[62]. A population weighted monoexponential function was used to model the CODEX decay observed in the citrate free CitApc sample with 30% population in the bound state, with the fitted spin diffusion rate $k$ of 0.0053 from the citrate bound CitApc sample.

## Reporting summary

Further information on research design is available in the Nature Portfolio Reporting Summary linked to this article.

## Data availability

X-ray data of the Gt PASp domain in citrate bound and citrate free forms determined in this work are available at protein data bank (PDB), with the accession numbers: 8BGB (Gt PASc WT), 8BIY (R93A mutant) and 8BJP (N288D mutant). PDB codes of previously published structures used in this study are 2J80 (citrate-bound periplasmic domain of CitA) and 5FQ1 (cytoplasmic PAS domain of CitA). NMR assignment data of the Gt CitApc in citrate bound and citrate free forms, the Gt PASc WT and N288D mutant are available at biological magnetic resonance data bank (BMRB), with the accession numbers: 51759, 51760, 51764, and 51765. Raw NMR data generated for this study have been deposited in the data repository Edmond under the accession code EPU9PQ https://doi.org/10.17617/3.EPU9PQ. Source Data are provided as a Source Data file. Source data are provided with this paper.

## Code availability

The code used for CODEX analysis is as published in Somberg et al.[65]. The custom written code used for COLD experiments has been published in Weisenburger et al.[26]

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

## Acknowledgements
We acknowledge discussions with Dariush Hinderberger on EPR results. We thank the beamline staff at SLS, X10SA for support with x-ray data collection. The authors also acknowledge funding by the Max Planck Society (C.G. and V.S.), as well as from the Deutsche Forschungsgemeinschaft with the following grants: Emmy Noether AN1316/1-1 (L.B.A.), UN 49/21-1 (G.U.), and GR 1211/18-1 (C.G.).

## Author contributions
Conceptualization: C.G., L.B.A., and S.B. Methodology: C.G., L.B.A., S.B., V.S., and G.U. Investigation: X.C.Z., K.X., D.B., M.Salvi., K.G., J.M., S.W., M.Stopp., and B.S. Funding acquisition: C.G., L.B.A., V.S., and G.U. Project administration: C.G., L.B.A., and S.B. Supervision: C.G., L.B.A., S.B., V.S., and G.U. Writing – original draft: X.C.Z. Writing – review & editing: all

## Funding

## Competing interests
The authors declare no competing interests.
