## [Transparent Peer Review file · Nature Communications]

Mechanism of sensor kinase CitA transmembrane signaling

Corresponding Author: Professor Christian Griesinger

Version 0:

Reviewer comments:

Reviewer #1

(Remarks to the Author)

This manuscript of Griesinger and Andreas is a true tour de force, showing a fine selection of advanced structural biology methods for the investigation the transmembrane signaling mechanism of a citrate-sensing membrane-embedded Histidine Kinase called CitA. This manuscript seeks to establish an atomistic understanding of the structural remodeling of the entire Kinase in a near-native membrane environment. This is a truly impressive study.

The marquee finding of this copious study is the structural rearrangement of the cytoplasmic PASc domain from an antiparallel to a parallel conformation, induced by the binding of citrate to the periplasmic PASp domain. This is demonstrated by two very elegant techniques - cryogenic optical localization (COLD) and DNP-enhanced ¹⁹F solid-state NMR using the CODEX experiment. I am particularly impressed by the later approach, which is a creative, novel application of ¹⁹F ssNMR.

I have a few comments.

- 1) The piston-like formation of an extra-helical turn for residues 154-157 upon citrate binding is intriguing. This region features two anionic residues (E156, D157) directly at the lipid-headgroup region, which suggests an impact of the membrane composition. Have the authors investigated the impact of the membrane composition on the structuring of the PASp/TM2-linker? Note that this situation appears somewhat reminiscent to the to the formation of an extra-helical turn in the KcsA-Kv1.3 channel (<https://www.pnas.org/doi/10.1073/pnas.1305563110>). There, the formation of the extra-helical turn is induced by the presence of zwitterionic lipids.
- 2) The lipid / water accessibility (especially what is discussed for residues 180 - 185) cannot be readily gauged from the provided data (Figure S4). Could the authors please show a convincing piece of evidence for differential lipid accessibilities for residues 180 – 185?
- 3) Based on Figure S4, it appears that the spectral quality is much better for the bound state. Could the authors please comment on this – if the spectral quality is compromised for the free state, then how challenging is the analysis for this state?
- 4) Could the authors find a different illustration that shows where the additional helical structuring occurs upon target binding? I am afraid this is not well conveyed by the current illustration, where residues 154-157 are helical in both states. This is not the messages that the authors like to spread. Please also indicate the citrate binding site. Also, in Figure 2B, green residues are not visible.
- 5) I would appreciate if the authors could provide detailed experimental settings for their ¹⁹F ssNMR experiments, including the T1 of their ¹⁹F samples.

Reviewer #2

(Remarks to the Author)

The manuscript entitled "Mechanism of Sensor Kinase CitA Transmembrane Signaling" describes the structural mechanism by which the membrane-bound histidine kinase, CitA, senses extracellular citrate to activate intracellular phosphorylation signaling pathways. CitA interacts with citrate at the periplasmic PASp domain, inducing structural changes in the cytoplasmic region to activate intracellular signaling. The authors used ¹H-detected solid-state NMR spectroscopy to find

changes in the secondary structure of CitApc, which contains the essential elements in the full-length CitA for signaling, upon binding to citrate at the PASp domain. The authors then conducted X-ray crystallographic analyses to determine the three-dimensional structures of the isolated cytoplasmic PASc domains of both the wild-type and its N288D variant. Interestingly, the wild-type and N288D variants of the PASc domains exhibited two different dimeric configurations, where the wild-type PASc domain assumes a parallel orientation, and the N288D variant assumes an anti-parallel helix orientation at the dimeric interface. The authors further compared the homodimers formed by these two PASc domain constructs using solution NMR and showed that there exists a two-state chemical exchange at the dimer interface. The structural change accompanying the distance change in the PASc dimer was also confirmed in CitApc upon interaction with citrate.

Overall, the manuscript presents a combination of sophisticated methods, including NMR, X-ray, and fluorescence spectroscopy, and seems to be of interest to the readership of Nature Communications. However, the authors should address the following concerns before publication.

Major points:

1. The N288D mutation is introduced abruptly in the manuscript, without any detailed description regarding where the residue is located and why it could modulate dimer formation. The authors should provide sufficient explanation for this mutation.
2. The CEST experiment appears to be the sole piece of evidence for the statement: "this indicates a dimer interface rearrangement of the PASc domain in isolation where the PASc domains most likely switch between the parallel and anti-parallel dimer forms." However, the CEST profiles of E215 from the wild-type and N288D variants (Fig. 2E) show virtually identical curves, indicating that the major and minor states possess the same chemical shifts with the same population. Conversely, the X-ray structures and solvent PRE results suggest that the wild type assumes the parallel orientation, and the N288D variant assumes the anti-parallel dimers as the major states, respectively. The authors should elaborate on this point.
3. Figure 4 indicates that the structural change of the PASc dimer from parallel to anti-parallel configurations is connected to changes in enzymatic activity. However, I don't see any mechanistic rationale for this point in the manuscript. The authors should elaborate on this point in the manuscript, explaining how the change in the PASc dimer is related to the changes in enzymatic activity.

Minor points:

1. In the statement "Furthermore, a loosened 192KK193 linker would explain the faster dynamics in the PASc domain in the citrate-bound state, indicated by the lower visibility of the PASc domain in cross-polarization (CP) based solid-state NMR spectra of CitApc with both ^{13}C and ^1H detection" (line 115), the authors' meaning of "lower visibility" is unclear. They should clarify this term.
2. In the COLD experiment (Fig. 2D), what is the error estimation value for the presented 41.3\AA for the N288D variant? Is this value statistically significantly different from the 9.0\AA value for the wild-type? The data for the wild-type should also be included.
3. In Fig. 1F, the authors present the secondary structure model derived from the chemical shift. The figure is difficult to comprehend as it is shown in a cartoon representation, and the labels of the residues are unclear regarding where they point. It would probably be better to show the CA or N atoms to indicate where each residue is located. Fig. 1D should be similarly labeled to clearly present the location of each residue in the structure.
4. In Fig. 1E, some of the labels at the bottom are garbled, and this should be corrected.
5. In Fig. 2A, the entire structure should be shown, as in Fig. 2B. Currently, the figure is trimmed at the edge of the panel, and this should be adjusted.
6. In Fig. 2A and 2B, the residues that exhibited CEST profiles are colored red. However, this information should be distinguished from the structure determination and findings of parallel and anti-parallel orientations. The results of the CEST experiment should be presented in independent panels, where residue labels are paired with the corresponding CEST profiles.

Reviewer #3

(Remarks to the Author)

The manuscript "Mechanism of sensor kinase CitA transmembrane signaling" by Zhang et al. is devoted to the studies of transmembrane signaling by PAS domain of sensory histidine kinases (HKs) of a two-component signaling system (TCS). TCS is the most abundant signaling system in nature and HKs are the main sensors of TCS. For many years the mechanism of transmembrane signaling by the sensors was a challenge. In 2017 on the basis of the structures of inactive and active states of NarQ HK solved to high resolution the mechanism of the sensors comprising HAMP domains (HAMP-HKs) was proposed (Guschin et al., Science 2017). However, the mechanism of TM signaling of PAS-HK remained unknown. Zhang et al. used PASp-TM-PASc truncation of the full-length sensor citrate kinase (CitA) of TCS for NMR studies of TM signaling mechanism. In addition, they solved the X-ray crystallographic structures of extracellular PASp and intracellular PASc domains and found out that PASc domain reveals both parallel and anti-parallel dimer conformations with the distances between C-terms of the protomers 7 and 40\AA correspondingly. Their NMR study of PASp-TM-PASc construct embedded into a lipid bilayer showed the corresponding distances equal to 13.6 ± 3 and $\geq 20\text{\AA}$. They also found out that "an inter-N308C distance of 41.3\AA was obtained in the N288D mutant PASc, which agrees with the expected distance in the anti-parallel dimer arrangement (Figure 2C and D). Regarding WT PASc, an inter-N308C distance of 9.0\AA is observed, which fits the expected distance based on the crystal structure of the parallel dimer." Combining all the data, the authors state that the TM signaling results in anti-parallel to parallel PASc transition. This is a central point of the proposed mechanism.

Taking into account a great importance of TCS and the presence of PAS domains in about 30% of HKs and the fact that the authors proposed a new mechanism of PAS involved in TM signaling, I would potentially support this work for publication. Nevertheless, I would suggest a revision of the manuscript before it can be published.

First, crystal structures show the distances 7 and 40 Å but NMR data give quite a different estimate of 13.6 ± 3 and ≥ 20 Å. Moreover, the numbers 13.6 ± 3 and 20 Å are quite close if we take into account the mathematical definition of the errors equal to ± 3 Å. Therefore, strictly speaking these data cannot be used as proof of the working hypothesis. The authors should provide a more careful discussion of the relevance of these data to the proposed mechanism. Anyway, these data cannot be considered as proof of the mechanism. In general, I would rather consider that the authors proposed a new well supported hypothesis of the transmembrane signaling (and this is very valuable) than a solid proof of the mechanism. Nevertheless, the work described in the manuscript is a significant step forward to the understanding of transmembrane signaling by PAS HKs.

One more comment. It is also desirable to discuss more comprehensively the applicability of the proposed mechanisms to other PAS containing HKs.

There are some minor comments:

Abstract. Please, say at the beginning what is not known about the mechanism of TM signaling.

Short summary. In your work you discuss PAS HKs. The mechanism is not known for this type of HKs. The mechanism of TM signaling has already been proposed for HAMP HKs. Please, correct that you mean PAS HKs.

Introduction. First paragraph. A general Supplementary Figure illustrating the modular structure of HKs is desirable.

Line 85. Could you explain briefly why you used R93A mutant.

Line 125. *E. Coli* should be in italics.

Lines 198-199. You had better speak in terms of a hydrophobic mismatch.

Line 267. DMPC membranes have a considerably thinner bilayer in comparison with the most biological membranes. Please, discuss how this could influence your NMR data.

Line 366. "proteins" should be replaced with "peptides".

Reviewer #4

(Remarks to the Author)

Please see attached reviewer comments.

Version 1:

Reviewer comments:

Reviewer #1

(Remarks to the Author)

I thank the authors for considering my comments, and I congratulate them to their work. This seems good progress towards a final manuscript.

However, I am afraid I am still not convinced by the analysis of the water and lipid accessibility of free and bound states. Given that the spectral quality of the free state is challenging, I would like to see convincing data how the 'water and lipid intensities' were derived for both states. I would like to ask the authors to show 1D cross-section for the signals that are discussed in Figure S6B.

Minor comment: If the anionic residues 156-157 were indeed modulated by zwitterionic lipid headgroups, did the authors observed spectral differences in anionic lipids? If the authors did not resolved ssNMR chemical shifts for these two residues in other lipid environments, I would appreciate if the authors made the reader aware that these residues could be modulated by the membrane composition, something that could be different in membranes that resemble the native composition more closely.

Reviewer #2

(Remarks to the Author)

The revised manuscript by Zhang et al. has appropriately addressed the concerns I raised regarding the original manuscript. Therefore, I now recommend the publication of this work.

Reviewer #3

(Remarks to the Author)

Reviewer #4

(Remarks to the Author)

The authors have adequately addressed all of my concerns. In particular, I appreciate the addition of the WT data in Figure 2 and the more lengthy description of the fitting for the COLD histograms. The manuscript, specifically with regards to the COLD microscopy my review focuses on, is suitable for publication as is.

Version 2:

Reviewer comments:

Reviewer #1

(Remarks to the Author)

I thank the authors for the additional clarifications and small adjustments. This is a very strong study to which I like to congratulate the authors!

Response to Reviewers

The authors thank the reviewers and editors for their work and suggestions. In the following point to point response, the authors' replies are colored in blue.

Reviewer 1

1. Comment: The piston-like formation of an extra-helical turn for residues 154-157 upon citrate binding is intriguing. This region features two anionic residues (E156, D157) directly at the lipid-headgroup region, which suggests an impact of the membrane composition. Have the authors investigated the impact of the membrane composition on the structuring of the PASp/TM2-linker? Note that this situation appears somewhat reminiscent to the to the formation of an extra-helical turn in the KcsA-Kv1.3 channel (<https://www.pnas.org/doi/10.1073/pnas.1305563110>). There, the formation of the extra-helical turn is induced by the presence of zwitterionic lipids.

Reply: Indeed, the residues 156-157 are exactly at the lipid head group region. Our first priority was to investigate the impact of the different lipids (including different lipid headgroups) on the spectroscopic quality, due to the challenges CitApc posed for structural biology studies.

Next, we have extensively investigated if CitApc is still functional using ^{31}P NMR. From our study (Figure S13), we have found that in DMPC liposomes the CitA protein still retains its functionality. Therefore, we conclude that the lipid headgroup chosen in this study does not impair the functional structural elements crucial for CitA signaling. The lipids in this study at ambient temperature resemble the ones of thermophilic bacteria regarding charge and mobility of the chain at 60°C (see reference: M.M. Donato, A.S. Jurado, M.C. Antunes-Madeira, V.M.C. Madeira. Membrane Lipid Composition of *Bacillus stearothermophilus* as Affected by Lipophilic Environment Pollutants: An Approach to Membrane Toxicity Assessment. *Arch. Environ. Contam. Toxicol.* 39, 145-153 (2000).). Yet, we need to use a thermophilic protein at ambient to reduce dynamics for optimal NMR detection.

2. Comment: The lipid / water accessibility (especially what is discussed for residues 180 - 185) cannot be readily gauged from the provided data (Figure S4). Could the authors please show a convincing piece of evidence for differential lipid accessibilities for residues 180 – 185?

Reply: Thank you for bringing this to our attention. We have added a figure displaying the changing intensity of these lipid contacting residues (new Figure S6B, shown below).

Figure. S6. Differential lipid contacts of CitA TM residues. (A) Lipid contacting residues in the TM helices mapped onto the CitA topology map in the free (left, blue) and bound (right, yellow) state of CitA. (B) Peak signal to noise ratio of residues G180 to V191 in the (H)HNH lipid contacting spectra in the free (blue) and bound (orange) state. Residues A184 and V185 show lipid contacts in the bound state but not in the free state. Residue A190 loses its lipid contact upon citrate binding. Peaks with signal to noise ratio larger than 3 (dashed grey line) are considered lipid contacting. The border of lipid contact in the TM1 helix does not change while the border of lipid contact in the TM2 helix is shifted consistent with additional C-terminal residues entering the membrane.

3. Comment: Based on Figure S4, it appears that the spectral quality is much better for the bound state. Could the authors please comment on this – if the spectral quality is compromised for the free state, then how challenging is the analysis for this state?

Reply: Yes. This difference in spectral quality was in fact already observed in previous work (reference 6).

For PAsp this is easily explained since citrate stabilizes its secondary structure and fold and therefore reduces its dynamics. This leads to better spectral dispersion and narrower line widths of the resonances.

In PAsc, more peaks above the detection limit are there in the citrate free state, which makes assignment of the PAsc in the citrate free state easier. This can be seen, for example, by the peak belonging to the residue A287, which is only present in the citrate free state.

Nevertheless, the assignment is still challenging due to the sheer size of the protein. One can gauge the difficulties by the amount of measurement time as well as the percentage of assignment (Tables S1-S4).

4. Comment: Could the authors find a different illustration that shows where the additional helical structuring occurs upon target binding? I am afraid this is not well conveyed by the current illustration, where residues 154-157 are helical in both states. This is not the

messages that the authors like to spread. Please also indicate the citrate binding site. Also, in Figure 2B, green residues are not visible.

Reply: We apologize for the misleading topology illustration. Changes have been made to the figure 1 according to the reviewer suggestions. 154-157 are no longer shown as a helix.

The residues constituting the citrate binding site have been colored red in figure 1B and highlighted. The most important binding-pocket-forming residues are V85, H96, S105, M106, I107 and G108.

In Figure 2B, green arrows now point to the C-terminal residues where the dyes have been introduced.

5. Comment: I would appreciate if the authors could provide detailed experimental settings for their ^{19}F ssNMR experiments, including the T1 of their ^{19}F samples.

Reply: Details of the experimental setup have been added in the material and methods section. T1 can be estimated directly from the S0 of the CODEX experiment and is roughly 320 ms for the citrate free ^{19}F sample (Figure S12).

Reviewer 2

Major points:

1. Comment: The N288D mutation is introduced abruptly in the manuscript, without any detailed description regarding where the residue is located and why it could modulate dimer formation. The authors should provide sufficient explanation for this mutation.

Reply: To clarify why the N288D mutant was chosen, we added the following sentence:

“We first generated functional mutants in the PASc domain from *G. thermodenitrificans* CitA, based on functional mutation loci in *E. coli* DcuS (Ref. 26) identified through systematic mutagenesis and based on sequence alignment between *E. coli* DcuS and *G. thermodenitrificans* CitA. The N288D CitA was derived from N304D of DcuS which kept DcuS in the OFF state (Ref. 26). When the function of the N288D CitA mutant was assessed by a β -galactosidase assay performed in *E. coli* CitA reporter strains (Figure 2F) we found that N288D CitA, unlike N304D of DcuS, was fully functional, i.e. activated by citrate.”

2. Comment: The CEST experiment appears to be the sole piece of evidence for the statement: “this indicates a dimer interface rearrangement of the PASc domain in isolation where the PASc domains most likely switch between the parallel and anti-parallel dimer forms.” However, the CEST profiles of E215 from the wild-type and N288D variants (Fig. 2E) show virtually identical curves, indicating that the major and minor states possess the same chemical shifts with the same population. Conversely, the X-ray structures and solvent PRE results suggest that the wild type assumes the parallel orientation, and the N288D variant assumes the anti-parallel dimers as the major states, respectively. The authors should elaborate on this point.

Reply: We thank the reviewer for the possibility to clarify the findings. The E215 CEST profiles show in both cases the major conformation at around 114 ppm and the minor conformation at around 120 ppm. That this amino acid shows the same chemical shift for the major states despite the fact that these major states are parallel (wt) and antiparallel (mutant) can indeed be the case. That a second state is also present is clearly visible from the CEST profiles. The identity of this alternative state cannot be deduced from the chemical shifts, yet, given the fact that the PAS domains crystallize in the two states make the presence of the other state probable.

Further evidence for these two forms existing in solution comes from the COLD experiment and MINFLUX (ref.29). In COLD, the wt shows distances between the two C-terminal C308 residues compatible with the parallel dimer while the distance of the C308 residues in the mutant conforms to the antiparallel dimer. The same is found in MINFLUX

for the wt that showed however two distances, compatible with interconversion between parallel and antiparallel dimer arrangement.

We have included the following sentences in the CEST paragraph:

“Transition between conformations of the isolated PASC domain is observed by chemical exchange saturation transfer (CEST), which detects lowly populated states when they undergo millisecond (ms) timescale conformational exchange with the main population³⁰. For the PASC domain in CitA, this CEST effect was observed mainly in the residues from the N-terminal helices that form the dimer interface in both the anti-parallel and parallel arrangements (Figure 2E and S8). This indicates a dimer interface rearrangement of the PASC domain in isolation where the PASC domains switch most probably between the parallel and anti-parallel dimer forms, although CEST does not reveal the identity of the conformations exchanging. Although this dimer exchange seen in the WT PASC is functionally more relevant, its occurrence in the N288D PASC mutant helps to rationalize why this point mutation does not disturb the functionality of the full length CitA (Figure 2F). MINFLUX²⁹ of the PASC domain confirms the conformation switching of wt. Still, connecting a PASC domain conformation with a specific citrate receptor state requires a membrane embedded construct.”

3. Comment: Figure 4 indicates that the structural change of the PASC dimer from parallel to anti-parallel configurations is connected to changes in enzymatic activity. However, I don't see any mechanistic rationale for this point in the manuscript. The authors should elaborate on this point in the manuscript, explaining how the change in the PASC dimer is related to the changes in enzymatic activity.

Reply: We had intentionally refrained from elaborating on this, since this work has largely been done without the kinase domain. We do speculate that the change in the C-terminal inter-dimer distance is the key to controlling the rate of cross-phosphorylation of the kinase domains. This is suggested to be the mechanism of activity regulation, however, it remains speculative.

We have included the following sentence in the Conclusion:

“Mechanistically, these data suggest that the kinase core domains are held apart due to the 45 Å distance separating the C-termini of PASC in the citrate-free state, while they are brought together to switch on cross-phosphorylation in the citrate-bound state, based on the 11 Å distance between the C-termini of PASC in the citrate-bound state.”

Minor points:

1. Comment: In the statement “Furthermore, a loosened 192KK193 linker would explain the faster dynamics in the PASC domain in the citrate-bound state, indicated by the lower visibility of the PASC domain in cross-polarization (CP) based solid-state NMR spectra of

CitApc with both ^{13}C and ^1H detection” (line 115), the authors' meaning of “lower visibility” is unclear. They should clarify this term.

Reply: Throughout the manuscript, “visibility” has been changed to “assignment completeness or peaks above the detection limit”, (case specific), for clarity.

2. Comment: In the COLD experiment (Fig. 2D), what is the error estimation value for the presented 41.3\AA for the N288D variant? Is this value statistically significantly different from the 9.0\AA value for the wild-type? The data for the wild-type should also be included.

Reply: The uncertainty for the distance value is $\pm 3\text{\AA}$. The value of 41.3\AA for the N288D variant is statistically significantly different from the 9.0\AA of the wild type. We have added a reproduction of Figure 1g from Ref. 23 (Weisenburger et al., Nature Methods 2017) to include the data for the wild type in Figure 2D.

3. Comment: In Fig. 1F, the authors present the secondary structure model derived from the chemical shift. The figure is difficult to comprehend as it is shown in a cartoon representation, and the labels of the residues are unclear regarding where they point. It would probably be better to show the CA or N atoms to indicate where each residue is located. Fig. 1D should be similarly labeled to clearly present the location of each residue in the structure.

Reply: Figure 1F has been edited accordingly. The side chains are now shown and the label placed around the side chain for clarity. Similar changes are done for figure 1D, wherein the side chains of the residues L154 and I158 are shown to make the position of the residues of interest clear.

4. Comment: In Fig. 1E, some of the labels at the bottom are garbled, and this should be corrected.

The authors thank the reviewer for catching this file conversion mistake. It is now corrected

5. Comment: In Fig. 2A, the entire structure should be shown, as in Fig. 2B. Currently, the figure is trimmed at the edge of the panel, and this should be adjusted.

Reply: Panel A in Figure 2 has been adjusted in the figure to show the entire structure.

6. Comment: In Fig. 2A and 2B, the residues that exhibited CEST profiles are colored red. However, this information should be distinguished from the structure determination and findings of parallel and anti-parallel orientations. The results of the CEST experiment should be presented in independent panels, where residue labels are paired with the corresponding CEST profiles.

Reply: We prefer to keep the information from the crystal structures, since it helps to place the residues in the context of the determined structure.

To give more clarity, additional labels have been added to residues that show CEST exchange in the N288D mutant and in WT PAsC, respectively, in figure 2A and 2B.

Reviewer 3

1. Comment: First, crystal structures show the distances 7 and 40 Å but NMR data give quite a different estimate of 13.6 \pm 3 and \geq 20 Å. Moreover, the numbers 13.6 \pm 3 and 20 Å are quite close if we take into account the mathematical definition of the errors equal to \pm 3 Å. Therefore, strictly speaking these data cannot be used as proof of the working hypothesis. The authors should provide a more careful discussion of the relevance of these data to the proposed mechanism. Anyway, these data cannot be considered as proof of the mechanism. In general, I would rather consider that the authors proposed a new well supported hypothesis of the transmembrane signaling (and this is very valuable) than a solid proof of the mechanism. Nevertheless, the work described in the manuscript is a significant step forward to the understanding of transmembrane signaling by PAS HKs.

Reply: Regarding significance of the distance determined through CODEX, we would like to point to the comparison to simulated distance shown as grey curves in Figure 3A. By comparing the decay curve of our data to simulated curves with known distance, it is clear that a significant inter-dimer distance change occurs at the C-terminal end of PASc.

From this statistically significant difference of the CODEX signals in the ON and OFF state we conclude that a structural change is definitely happening with statistical significance. The exact distances from the X-ray structure and COLD of the isolated PASc domains and CODEX from CitApc have some variation due to the fact that the labels are different. In X-ray we measured the distance between the C α atoms, in COLD a big dye is attached whose exact position with respect to the tagged residue C308 is not known. Thus, finding an obvious difference in the CODEX profiles indicates that different structures must prevail. The assignment to parallel and antiparallel is the most probable one (Occam's razor) taking into account that the isolated PASc domains adopt the parallel and antiparallel orientation. We have already formulated the language in such a way that the reader does not get the impression of an unambiguous proof but rather a well supported hypothesis.

Comment: One more comment. It is also desirable to discuss more comprehensively the applicability of the proposed mechanisms to other PAS containing HKs.

Reply: Based on crystal structures of the CitA PASp domain in ligand bound and free state from *Klebsiella pneumoniae* (ref 5) and GT in this work, signaling via a sensor PASp domain is now established as a small contraction of the core β -sheet upon ligand binding that is passed on to the cytoplasmic domains by pulling up the C-terminal transmembrane helix. It can be predicted that structurally very similar sensor domains like those of the HKs DcuS and PhoQ (ref 7.) should have a similar signaling mechanism.

Next, there are data from other histidine kinases that can be reconciled with rearrangement of their cytoplasmic PAS domains upon signaling similar to CitA in this work. For example,

Cys-scanning mutagenesis targeting the β -sheet of the cytoplasmic PAS domain of sensor kinase BvGs, resulted in changing patterns of S-S bond-mediated cross-linking dependent on the signaling state, in line with changing quaternary structure of the PAS domain dimer during signal transduction (ref 24). Also the finding that protein flexibility in the PAS domain of DcuS is involved in signal transduction of this HK (Etzkorn, Manuel, et al. "Plasticity of the PAS domain and a potential role for signal transduction in the histidine kinase DcuS." *Nature structural & molecular biology* 15.10 (2008): 1031-1039.) might imply rearrangement of these PAS domains during signal transduction. Further, replacing the oxygen-sensing PAS domain of the *B. japonicum* FixL kinase with a photosensory PAS domain (ref 33) resulted in a chimeric protein retaining the catalytic efficiency of FixL but responding to blue light instead of oxygen. This led the authors to conclude that a high degree of similarity in the mechanisms of the parent chemosensor and the chimeric photosensor is implied (ref 33).

Thus, the signal transduction mechanism elucidated for CitA might be applicable more generally to HKs with cytoplasmic PAS domains.

We introduced a paragraph at the end of the "results" section:

"While we investigated CitA from GT, there is some evidence that PAS domains of other sensory kinases switch dimer arrangements as well: Cys-scanning mutagenesis targeting the β -sheet of the cytoplasmic PAS domain of sensor kinase BvGs, resulted in changing patterns of S-S bond-mediated cross-linking dependent on the signaling state, in line with changing quaternary structure of the PAS domain dimer during signal transduction²⁴. Replacing the oxygen-sensing PAS domain of the *B. japonicum* FixL kinase with a photosensory PAS domain³³ resulted in a chimeric protein retaining the catalytic efficiency of FixL but responding to blue light instead of oxygen. This led to the conclusion that a high degree of similarity in the mechanisms of the parent chemosensor and the chimeric photosensor is implied³³."

There are some minor comments:

1. Comment: Abstract. Please, say at the beginning what is not known about the mechanism of TM signaling.

Reply: The sentence "however, a uniform structural model is still missing for their transmembrane signaling mechanism." has been added.

2. Comment: Short summary. In your work you discuss PAS HKs. The mechanism is not known for this type of HKs. The mechanism of TM signaling has already been proposed for HAMP HKs. Please, correct that you mean PAS HKs.

Reply: Sentence from short summary lines 9 to 11 has been changed to “Despite decades-long efforts, the field currently lacks a precise model for the structural mechanism underlying signal transduction of HKs with extracytoplasmic and cytoplasmic PAS domains”.

3. Comment: Introduction. First paragraph. A general Supplementary Figure illustrating the modular structure of HKs is desirable.

Reply: A Figure S1 has been added with the modular structure of CitA.

4. Comment: Line 85. Could you explain briefly why you used R93A mutant.

Reply: The sentence “The R93A mutant is used to access the citrate free state, since the wild type cannot be produced without citrate due to very tight binding.” Has been added.

More details of the mutant is available in the citation from the previous sentence.

5. Comment: Line 125. *E. Coli* should be in italics.

Reply: The corresponding format change has been made.

6. Comment: Lines 198-199. You had better speak in terms of a hydrophobic mismatch.

Reply: Either one state or the other has a hydrophobic mismatch (unless this is relieved by a change in the helix tilt). Said sentence has been included the phrase “, which potentially induces a hydrophobic mismatch of the TM2 in one of the states unless this is relieved by a change in the helix tilt”.

7. Comment: Line 267. DMPC membranes have a considerably thinner bilayer in comparison with the most biological membranes. Please, discuss how this could influence your NMR data.

Reply: This question was discussed already in the reply to comment 1 of reviewer 1.

8. Comment: Line 366. “proteins” should be replaced with “peptides”.

Reply: This refers to the crystallization of PASC. The authors would respectfully disagree with this comment. These are proteins, not peptides. Therefore, said part has not been changed.

Reviewer 4

1. Comment: The authors make several references to the “PAS core.” It is unclear where the PAS core is in the structure and labelling it in Figure 1 would be helpful.

Reply: An additional comment on the PAS core has been added in the introduction. Namely, the explanation that “in the context of this paper, PAS core refers to the PAS domain without the N-terminal helix.” has been added.

2. Comment: Regarding the COLD data, no WT data for PASc is shown in this manuscript, yet it is discussed. I recommend that the relevant figures from reference 23, specifically 1g, should be reproduced in the main text adjacent to figure 2D or, in the least, shown in the SI.

Reply: We have added a reproduction of Figure 1g from Ref. 23 (Weisenburger et al., Nature Methods 2017) to include the data for the wild type in Figure 2C and 2D.

3. Comment: It is very unclear what the “model fit” is that produces the red curve in figure 2D. Can the authors add a sentence or two describing what is being fit here.

Reply: We have expanded the Figure caption for panel 2D in the manuscript to provide more clarity: “...by fitting a model function which is the convolution of a projection function and the Euclidian norm of a bivariate normal distribution of the two determined fluorophores positions.”

Furthermore, we have also added a more detailed description to the method section: “We graphically display distributions by histograms for all relevant data in Figure 2D. To extract quantitative information, we plot the histogram of measured distances together with a fit based on a Monte Carlo model and simulated annealing, also considering the localization precision values. In a simple two-dimensional distance measurement scenario, one expects the measured values to distribute symmetrically about a most probable value. In our case, the distances represent the projection of distances in a randomly oriented 3D arrangement, leading to an asymmetric distribution due to a cosine dependence. The final asymmetric distribution stems from the convolution of a projection function and the Euclidian norm of a bivariate normal Rice distribution to take into account the statistical error of each projection. Because of this skewed distribution, the expectation value does not coincide with the peak position. For the 3D reconstruction and orientation fitting, data sets above a threshold localization precision of 6.5 Å were excluded.”

4. Comment: In reference 23, the original COLD manuscript, the authors comment on the extraneous long-distance measurements that give values out to 90 angstroms. Here, in data that was presumably taken at a similar time to ref 23, these long-distance outliers are gone. Can the authors explain why there are no measurements above ~65 Angstroms in this data, but many in the WT? Was there an extra filtering step that occurred?

Reply: There was no extra filtering step applied, both datasets have been processed in the same way. This dataset did not show any distance values beyond approximately 65 Å.

Response to review:

Reviewer #1 (Remarks to the Author):

I thank the authors for considering my comments, and I congratulate them to their work. This seems good progress towards a final manuscript.

However, I am afraid I am still not convinced by the analysis of the water and lipid accessibility of free and bound states. Given that the spectral quality of the free state is challenging, I would like to see convincing data how the 'water and lipid intensities' were derived for both states. I would like to ask the authors to show 1D cross-section for the signals that are discussed in Figure S6B.

The reviewer is correct that these data are relatively noisy, which is why they are placed in the SI. The cross sections are included now, and the statement about them in the main text has been modified to make clear that the main purpose of the spectrum was confirmation of PAS vs TM assignments. The respective sentence reads: "Assignments in this region are confirmed by solvent transfer experiments, with lipid contact found for transmembrane residues, but absent for characteristic and well-separated resonances of the PAS domains²⁰ (Figures S5 and S6)"

Minor comment: If the anionic residues 156-157 were indeed modulated by zwitterionic lipid headgroups, did the authors observed spectral differences in anionic lipids? If the authors did not resolved ssNMR chemical shifts for these two residues in other lipid environments, I would appreciate if the authors made the reader aware that these residues could be modulated by the membrane composition, something that could be different in membranes that resemble the native composition more closely.

We have included the following sentence in the manuscript: "We note that the anionic residues 156 and 157 might be modulated by the membrane composition¹⁹⁻²¹, which is far more complex in native membranes. Here, our focus is on the structural response of CitApc to citrate. The relevance of the use of a simplified lipid bilayer is justified by the response of full-length CitA to citrate binding as observed via ³¹P NMR tracking of ATP hydrolysis (Fig. S10) when reconstituted in DMPC liposomes. "

In addition, we did some minor changes in the formulation.

In the manuscript “Mechanism of sensor kinase CitA transmembrane signaling,” the authors study a canonical histidine kinase, CitA, which is responsible for sensing extracellular citrate. Their central question is how small structural changes in the sensor domain upon citrate binding are transmitted and amplified across the membrane to result in robust signaling. The authors employ a litany of methods to investigate these conformational changes, including various forms of NMR, crystallography, and cryogenic fluorescence microscopy. Much of this is beyond this reviewer’s expertise and I have been instructed to comment specifically on the cryogenic light microscopy measurements. The cryogenic light microscopy experiments are employed to support the hypothesis that there is a dimer of the cytosolic domain, PASc, that is either in the parallel (free) or anti-parallel (citrate bound) conformation. The cryogenic light microscopy approach uses covalent labelling of the C terminus of PASc with Atto647N. Under liquid helium temperatures Atto647N stochastically blinks and provides many fluorescent photons before photobleaching allowing for very precise localization information and accurate distance measurements between the two dyes on the two C termini of the dimers. These distance measurements are performed in a mutant, N288D, which is shown by crystallography, to exist in the anti-parallel conformation. The distance measurements reveal an increased separation between the dyes, 41.3 Angstroms, compared to wild-type, 9 Angstroms. The work is well done and should be published. I have only a few minor comments to improve readability and clarify aspects of the work.

Minor Issues:

- 1.) The authors make several references to the “PAS core.” It is unclear where the PAS core is in the structure and labelling it in Figure 1 would be helpful.
- 2.) Regarding the COLD data, no WT data for PASc is shown in this manuscript, yet it is discussed. I recommend that the relevant figures from reference 23, specifically 1g, should be reproduced in the main text adjacent to figure 2D or, in the least, shown in the SI.
- 3.) It is very unclear what the “model fit” is that produces the red curve in figure 2D. Can the authors add a sentence or two describing what is being fit here.
- 4.) In reference 23, the original COLD manuscript, the authors comment on the extraneous long-distance measurements that give values out to 90 angstroms. Here, in data that was presumably taken at a similar time to ref 23, these long-distance outliers are gone. Can the authors explain why there are no measurements above ~65 Angstroms in this data, but many in the WT? Was there an extra filtering step that occurred?